# Improve Temporal Awareness of LLMs for Domain-general Sequential Recommendation

**Zhendong Chu** [1]  **Zichao Wang** [2]  **Ruiyi Zhang** [2]  **Yangfeng Ji** [1]  **Hongning Wang** [1]  **Tong Sun** [2]

## Abstract

Large language models (LLMs) have demonstrated impressive zero-shot abilities in solving a wide range of general-purpose tasks. However, it is empirically found that LLMs fall short in recognizing and utilizing *temporal* information, rendering poor performance in tasks that require an understanding of sequential data, such as *sequential recommendation*. In this paper, we aim to improve temporal awareness of LLMs by designing a principled prompting framework. Specifically, we propose three prompting strategies to exploit temporal information within historical interactions for LLM-based sequential recommendation. Besides, we emulate *divergent thinking* by aggregating LLM ranking results derived from these strategies. Evaluations on MovieLens-1M and Amazon Review datasets indicate that our proposed method significantly enhances the zero-shot capabilities of LLMs in sequential recommendation tasks.

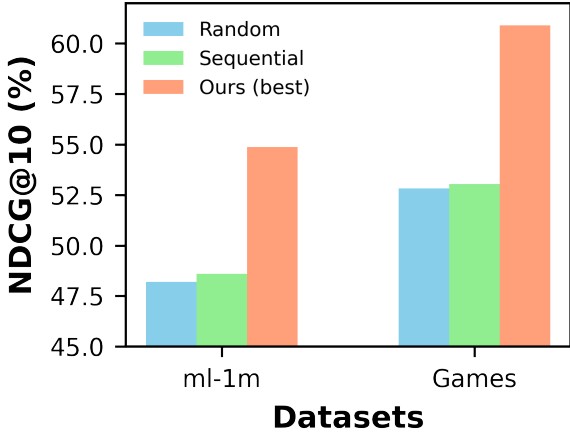

Figure 1: LLM-based sequential recommendation baselines show comparable performance even when historical interactions (Sequential) order is randomized (Random). `Tempura` significantly boosts performance by utilizing historical orders, *i.e.*, temporal information.

## 1. Introduction

Large language models (LLMs) such as ones with commercially available APIs including ChatGPT (Achiam et al., 2023) and Claude[1] have emerged as one of the primary, if not the de facto, choices in a wide range of applications thanks to their remarkable capabilities in dealing with natural language and generalizing to various domains without further fine-tuning. In deed, an emerging trend is to use natural language as a uniform interface and leverage the LLMs to complete a task.

Following this trend, recent research has been exploring the use of LLMs for processing sequential data, with applications such as sequential recommendation (SRS) (Hou et al., 2023b; Bao et al., 2023), which require LLMs to comprehend temporal patterns within user historical interactions. In the case of sequential movie recommendation, historical interactions such as users' movie watching records can be represented as natural language (i.e., movie titles and other meta data) for the LLMs to process and recommend the next movie, instead of item identifiers which are typically used in traditional recommender systems (Kang & McAuley, 2018; Sun et al., 2019). The extensive generalization ability and vast world knowledge (Wang et al., 2020; Singhal et al., 2023) of LLMs endow them with the potential to serve as a single model for many recommendation domains without fine-tuning, making it a general, capable, and easy-to-use alternative to traditional recommender systems that usually specialize in one selected domain and require extensive training or fine-tuning.

However, recent research shows that LLMs exhibit a limited sensitivity to temporal information in the input text, particularly in discerning changes in user interests (Hou et al., 2023b). In Figure 1, we compare the recommenda-

---

[1]University of Virginia, Charlottesville, VA, USA [2]Adobe Research, San Jose, CA, USA. Correspondence to: Zhendong Chu <zc9uy@virginia.edu>, Zichao Wang <jackwa@adobe.com>.

*Proceedings of the 1st Workshop on In-Context Learning at the 41st International Conference on Machine Learning*, Vienna, Austria. 2024. Copyright 2024 by the author(s).

[1]https://www.anthropic.com/index/claude-2

tion performance of LLM-based methods using randomized (denoted as **Random**) versus correctly ordered (denoted as **Sequential**) historical interactions on two widely-used SRS datasets. Both methods show similar performance, suggesting that LLMs are not effectively utilizing the temporal information present in the input text. This limitation stems from a lack of specialized mechanisms within LLMs to automatically recognize and utilize temporal information, which is crucial for understanding the context and progression within the data.

In this paper, we focus on improving LLMs' awareness and interpretation of temporal information, particularly within the SRS scenario. Temporal information is ubiquitous in real-world applications, such as recommender systems (McAuley, 2022), intelligent document processing (Fischer, 2001) and financial market analysis (Tsay, 2005). By effectively capturing and integrating this temporal aspect, we have the opportunity to significantly enhance the understanding of user preferences via LLMs, thus providing users with better recommendations that suit their backgrounds, needs, and preferences. This improvement is also important for boosting the effectiveness of LLMs in downstream applications, where accurate user preference modeling is crucial (McAuley, 2022). To this end, we design a principled prompting framework, which is training-free and domain agnostic. We name our approach as **Tempura** (phonetically similar to *Temporal Prompt*). Our main contributions are:

- We propose a principled method to construct in-context examples (Min et al., 2022) for sequential recommendation, by analyzing how Transformer-based SRS models (e.g., Kang & McAuley (2018)) learn to utilize temporal information.
- Inspired by the results in neuroscience (Nobre & Van Ede, 2018; Griffiths et al., 1998), we add explicit structure analysis in input sequences as additional prompts, particularly temporal cluster analysis, to enhance the temporal understanding capabilities of LLMs.
- We emulate the process of *divergent thinking* (Runco, 1991) by aggregating ranking results derived from various prompting strategies.
- We evaluate our method on MovieLens-1M and Amazon Review datasets, the results show that our proposed method significantly enhances the zero-shot capabilities of LLMs in sequential recommendation tasks.

## 2. Related Works

**LLMs for recommendation.** Recently, the use of LLMs in recommendation systems has garnered significant research interest due to their capability to comprehend and encapsulate a user's preferences and past interactions through natural language (Fan et al., 2023; He et al., 2023). Current LLM-based recommender systems are primarily designed for rating prediction (Kang et al., 2023; Bao et al., 2023) and sequential recommendation tasks (Wang & Lim, 2023; Hou et al., 2023b; Xu et al., 2024). In both tasks, a user's previous interactions with items, along with other optional data like the user profile or item attributes, are concatenated to formulate a natural language prompt. This is then fed into an LLM with options for no fine-tuning (Wang & Lim, 2023), full-model fine-tuning (Chen, 2023) or parameter-efficient fine-tuning (Bao et al., 2023). Liu et al. (2023a) designs a series of prompts to evaluate ChatGPT's performance over five recommendation tasks. Wang et al. (2023) develops a ChatGPT-based agent to improve recommendation ability by using tools such as SQL and Web search. Contrary to existing works that focus on the tentative evaluation of LLMs' ability in recommendation, we focus on improving the LLM's inefficacy of utilizing temporal information by designing temporal-aware prompting strategies.

**Sequential recommendation.** Sequential recommendation (SRS) (Hidasi et al., 2015; Kang & McAuley, 2018) aims to predict the next interacted items based on historical interaction sequences. Early works follow the Markov assumption (Rendle et al., 2010), by designing various neural network models to capture user preference within interaction sequences, including Recurrent Neural Network (Hidasi et al., 2015; Li et al., 2017), Convolutional Neural Network (Tang & Wang, 2018), Transformer (Kang & McAuley, 2018; Sun et al., 2019), Graph Neural Network (Chang et al., 2021; Wu et al., 2019). However, most of these approaches are developed based on item IDs (Kang & McAuley, 2018) or attributes (Zhang et al., 2019) defined on specific domains, making it difficult to be generalized to other domains. Recently, Hou et al. (2023a), Hou et al. (2022) and Li et al. (2023) propose to learn unified item representations for SRS based on pretrained language models. They follow the paradigm that pretraining an unified text-based sequence encoder on source domains and then fine-tune the encoder on the target domain. However, all aforementioned methods need massive user interaction sequences on a specific domains and can not be easily transfer to unseen domains. In contrast, we propose utilizing LLMs to establish a domain-agnostic learning process for sequential recommendation systems. Our approach is training-free and readily generalizable to unseen domains using only prompts.

## 3. Methodology

In this section, we introduce Tempura in detail. As shown in Figure 2, Tempura consists of three major components: 1) a in-context learning module that learns sequential recommendation tasks from sequences of historical interactions; 2) a temporal structure analysis module that enhances the model's understanding by explicitly integrating cluster structures within the sequences; 3) a prompt ensemble

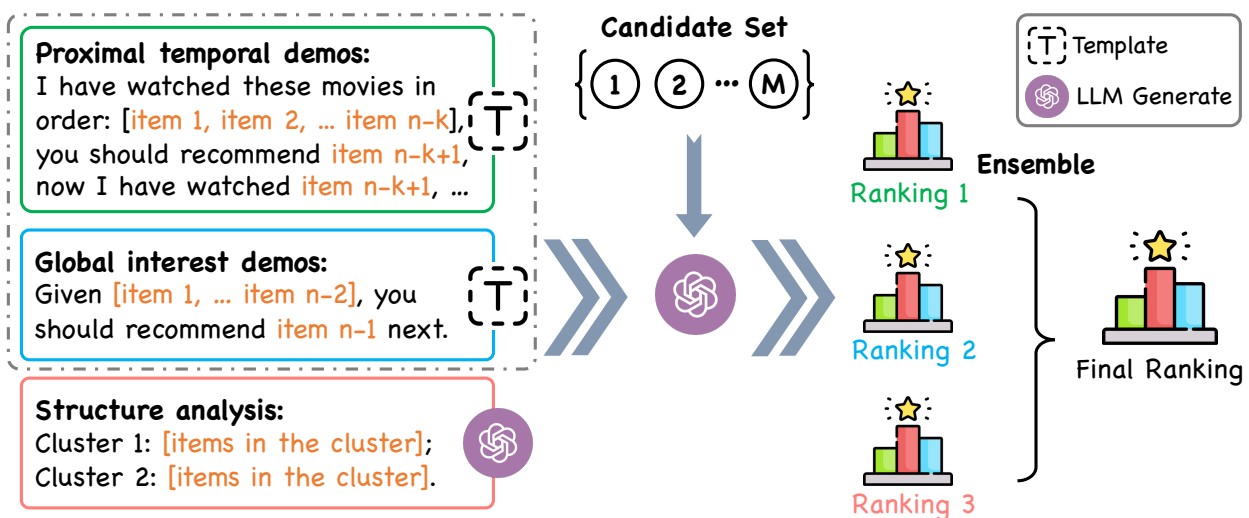

Figure 2: An illustrative overview of `Tempura`. We learn sequential recommendation via two kinds in-context demonstrations. Explicit cluster structure analysis is conducted to improve the temporal understanding capabilities of LLMs. Each prompting strategy independently generates a respective ranking by LLMs (marked by different colors). Rankings from different prompting strategies are aggregated to form the final ranking.

module that aggregates recommendation results from various prompting strategies. We begin with the definition of notations to be used in our technical discussions.

### 3.1. Problem Definition

Given a user's historical interactions $\mathcal{H} = \{i_j\}_{j=1}^n$, ordered chronologically up to timestamp $n$, the task of sequential recommendation involves ranking a set of *candidate* items $\mathcal{C} = \{i_j\}_{j=1}^m$ for the subsequent timestamp $n+1$. Items of higher interest are expected to be ranked at more prominent positions. In practice, candidate items are typically selected from the entire item set $\mathcal{I}$, where $m \ll |\mathcal{I}|$, through candidate generation models (Covington et al., 2016). Further, we follow the approach of Hou et al. (2022) by associating each item $i$ with a descriptive text $t_i$, which could be the item's name and its attributes or properties.

Different from training-based SRS models, we leverage general-purpose LLMs (e.g., ChatGPT) to solve the recommendation task in an instruction-following paradigm (Wei et al., 2021). Specifically, for each user, we construct a history prompt from the user's historical interactions $\mathcal{H}$, and a candidate item prompt from the candidate item set $\mathcal{C}$. The aforementioned prompts are concatenated along with an instruction that explicitly describes the recommendation task, forming the final prompt for LLMs. LLMs are anticipated to generate rankings of $\mathcal{C}$, reflecting user preferences, in accordance with the format specified by the instruction. A post-hoc text parser is employed to convert the natural language rankings generated by LLMs into structured ranked lists, which is used to calculate the ranking metrics (Hou

et al., 2023b).

### 3.2. Sequential Recommendation via In-Context Learning

Given the vast scale of LLMs, fine-tuning domain-specific models becomes impractical. Thus, we propose to learn sequential recommendation via in-context learning, offering a training-free approach that can be easily adapted across various domains by leveraging the world knowledge and comprehension capabilities of LLMs (Hou et al., 2023b; Harte et al., 2023). To this end, we first analyze the learning process of training-based SRS models, and then mapping it onto the principles of constructing effective in-context demonstrations.

The key distinction between SRS and other recommender systems lies in the SRS model's requirement to not only identify a user's preferences based on historical user-item interactions but also to track the evolution of the user's interests over time. Training-based SRSs depend on learning from large-scale user-item interaction data via GRUs (Hidasi et al., 2015) or Transformers (Sun et al., 2019). We utilize In-Context Learning (ICL) (Min et al., 2022) as a training-free alternative to learn a SRS model. We follow Dai et al. (2022) to analyze the learning process of training-based SRSs. Given the historical interaction sequence of an user, a trained Transformer-based SRS, such as SASRec (Kang & McAuley, 2018), can be represented as,

$$\mathcal{F}_{\text{SASRec}}(\boldsymbol{x}_n) = (W_0 + \Delta W)\boldsymbol{x}_n. \qquad (1)$$

where $W_0$ is the initialized parameter matrix, $\Delta W$ is the

update matrix and $\boldsymbol{x}_n$ is the representation of a candidate item. The output of $\mathcal{F}_{\text{SASRec}}$ is the score of the examined candidate item. In the back-propagation algorithm, $\Delta W$ is computed by accumulating the outer products of historic item representations $\boldsymbol{x}_i'^T$ and the error signals $\boldsymbol{e}_i$ of their corresponding outputs:

$$\Delta W = \sum_{i=1}^{n-1} \boldsymbol{e}_i \otimes \boldsymbol{x}_i', \tag{2}$$

where error signals $\boldsymbol{e}_i$ is the prediction error on the historic item $\boldsymbol{x}_i'$. Thus, the trained SASRec can also be rewritten into,

$$\begin{aligned}
\mathcal{F}_{\text{SASRec}}(\boldsymbol{x}_n) &= (W_0 + \Delta W)\boldsymbol{x}_n \\
&= W_0 \boldsymbol{x}_n + \sum_{i=1}^{n-1} (\boldsymbol{e}_i \otimes \boldsymbol{x}_i')\boldsymbol{x}_n \\
&= W_0 \boldsymbol{x}_n + \text{LinAtt}(E, X', \boldsymbol{x}_n),
\end{aligned}$$

where $\text{LinAtt}(V, K, \mathbf{q})$ denotes the linear attention operation, in which we regard error signals $E$ as values and interacted items $X'$ as keys, and the current input $\boldsymbol{x}_n$ as the query. The learning process of the SASRec model can be expained as the model predicting the next item in a sequence based on preceding items and updating itself based on the prediction error. The trained SASRec model is designed to update user preferences as the sequence expands, effectively tracking the evolution of the user's interests.

Let $\mathbf{q} = W_Q \boldsymbol{x}_n$ represent the attention query vector for the input candidate item $\boldsymbol{x}_n$. An ICL-based SRS can be represented as,

$$\mathcal{F}_{\text{ICL}}(\mathbf{q}) = (W_{\text{ZSL}} + \Delta W_{\text{ICL}})\mathbf{q}$$

where $W_{\text{ZSL}} = W_V X (W_K X)^T$ is the initialized parameters to be updated and $W_{\text{ZSL}}\mathbf{q}$ is the attention result in zero-shot learning (ZSL) setting, where no demonstration are given. $X$ denotes the input representations of query tokens before $\boldsymbol{x}_n$, such as the task description of sequential recommendation. Based on the results of Dai et al. (2022), the second term can be rewritten into,

$$\Delta W_{\text{ICL}}\mathbf{q} = \text{LinAtt}(W_V X', W_K X', \mathbf{q}),$$

where $X'$ denotes the input representations of demonstrations. Here we observe a similar form between $\mathcal{F}_{\text{SASRec}}$ and $\mathcal{F}_{\text{ICL}}$, where $W_V X'$ can be explained as the error signal from historic items. This analogy illustrates that by utilizing historic items as in-context demonstrations, an LLM can learn to capture the temporal information within the sequence of historical interactions. Hou et al. (2023b) discussed using the last item in the history as an in-context

demonstration. Based on our analysis, this method is equivalent to training the SASRec model solely with the last historical interaction, a practice insufficient for capturing the dynamic nature of historical interactions. Thus, we are motivated to use several historical interactions as demonstrations to improve the temporal awareness of LLMs.

**Proximal temporal demonstrations (PCL).** Based on the above principle, we design the following prompt to learn to capture temporal information via ICL,

> **Proximal temporal demonstrations**
>
> I have watched these movies in order: [item 1, item 2, ... item n-k], you should recommend item n-k+1, now I have watched item n-k+1, ...
> Now recommend a new movie to me.

Placeholders, highlighted in orange, structure the input for our model. The first placeholder captures the initial $n - k$ historic items, serving as the *context* for inferring user preferences. The subsequent placeholder is designated for the $n - k + 1$ item, illustrating the next item to be recommended based on the current context. Following this, we inform the LLM that the $n - k + 1$ item has been interacted with, indicating that the $n - k + 2$ item is the next recommendation target. This setup is repeated to create $k$-shot demonstrations. We utilize the most recent $k$ items as demonstrations to capture the proximal interest of the user. We denote this prompting strategy as PCL.

**Global interest demonstrations.** In previous studies (Kang & McAuley, 2018; Hou et al., 2023b), the number of historic items was constrained by the limited input length of models. Thus the whole interaction history is typically truncated and the most recent items are remained. Empirically, we also observed that extending the context window has limited impact on improving performance and may even detract from it. The reason could be: 1) the prolong context distract LLMs (Liu et al., 2023b); 2) too old history has little impact on the current user interest in the SRS scenario. However, simply omitting distant historic items risks overlooking users' long-term interests. Hence, we randomly sample a subset of historic items from the whole history sequence to retain user's global interest. Specifically, we use the same template as PCL, but the context is filled with randomly sampled historic items. Similarly, we incorporate the most recent items as in-context examples. We denote this prompting strategy as GCL.

### 3.3. Temporal Structure Analysis

It has been recognized in the neuroscience area that the human brain is more sensitive to temporal structures (Nobre & Van Ede, 2018; Griffiths et al., 1998) - "*Embedded relationships among the attributes of events over different timescales*

*carry predictions that guide proactive sensory and motor preparation in the brain*". Only providing item sequences may make it difficult for LLM to identify and utilize temporal patterns inside the sequence. Thus, we are motivated to explicitly provide temporal structures to LLM. Specifically, we conduct cluster analysis on the item sequence according to two criteria: items that are (1) *temporally proximate* and (2) *share similar features* should be clustered. In practice, we also use LLMs to complete the cluster tasks and find it can provide reasonable cluster results. The results are used as additional input to the LLM for ranking.

> **Structure analysis prompt**
>
> I have watched these movies in order: [item 1, item 2, ... item n]. Analyze the clusters within the history. Two criteria: 1) Similar items should be clustered together; 2) Temporal close similar items should be clustered.

### 3.4. Prompt Ensemble

The most straightforward way to combine various prompting strategies is to concatenate them and use the resulted long prompt. However, this approach risks exceeding the context length limitations of LLMs. Moreover, it has been observed that LLMs may lose important information within overly lengthy prompts (Liu et al., 2023b). To effectively utilize different prompting strategies, we propose ensembling the respective ranking outcomes derived from each. In this approach, we create several LLM sessions and obtain ranking lists with different prompts. Following Hou et al. (2023b), we explicitly define the output format for the ranking results produced by LLMs, and subsequently extract the ranking list using a post-hoc text parser. These ranking lists are aggregated to obtain the final ranking, as the process shown in Figure 2. Existing research also highlights the benefits of collaboration among multiple LLMs (Wu et al., 2023). Specifically, we assign scores to each rank in the ranking list. For instance, in a ranking list of 20 items, the item in the 1st place receives 20 points, the 2nd place item gets 19 points, and so on, decreasing by one point per rank. Finally, we sum the scores for each item across all rankings.

## 4. Experiments

In this section, to fully demonstrate the effectiveness of Tempura in improving temporal awareness of LLMs, we conduct a set of extensive experiments to study the following research questions: (1) Can Tempura improve LLM's performance on sequential recommendation compared to other methods? (2) Can Tempura enhance the sensitivity of LLMs to temporal information in the input data? (3) How do factors like history length, the number of in-context

examples or the choice of backbone LLMs influence the effectiveness of Tempura?

### 4.1. Setup

**Datasets.** The experiments are conducted on three widely-used public sequential recommendation datasets: (1) the movie rating dataset MovieLens-1M (Harper & Konstan, 2015) (**ML-1M**) where user rated movies are regarded as interactions, (2) one category from the Amazon Review dataset (Ni et al., 2019) named **Games** where reviews are regarded as interactions, and (3) another category from Amazon Review dataset named **Kindle**. We sort the interactions of each user by timestamp, with the oldest interactions first, to construct the corresponding interaction sequences. The movie or product titles are used as the descriptive text of an item.

**Evaluation configurations.** Following existing works (Kang & McAuley, 2018; Sun et al., 2019; Hou et al., 2023b), we apply the leave-one-out strategy for evaluation. For each interaction sequence, the last item is used as the ground-truth item. We adopt the widely used metric NDCG@N to evaluate the ranking performance over the given candidate set $\mathcal{C}$ where $N \leq |\mathcal{C}|$. In the remainder of this paper, unless otherwise specified, $|\mathcal{C}|$ is set to 20. The candidate set consists of one ground-truth item and 19 randomly sampled negative items.

**Baselines.** We consider three prompt-based baselines discussed in (Hou et al., 2023b): **Sequential prompting**: Arrange the historical interactions in chronological order. **Recency-focused prompting (RF)**: In addition to the sequential interaction records, a sentence is additionally added to emphasize the most recent interaction. **In-context learning (ICL)**: Similar to PCL, but only use the most recent historic item as the in-context example. We also consider three methods designed for domain generalization: **BM25** (Robertson et al., 2009) ranks items according to the textual similarity between candidates and historic items. **UniSRec** (Hou et al., 2022) equips textual item representations with an MoE-enhanced adaptor for domain fusion and adaptation. **VQ-Rec** (Hou et al., 2023a) learns vector-quantized item representations, which can map item text into a vector of discrete indices (i.e., item codes) and use them to retrieve item representations from a code embedding table in recommendations. Additionally, we report the results with each single prompting strategy, as well as the results from ensembling PCL and cluster analysis.

Training-based methods such as (Kang & McAuley, 2018; Sun et al., 2019) are not considered as baselines because: (1) They are designed based on item IDs, which can not be generalized to new domains with new ID spaces. (2) Our research focuses on improving the temporal awareness of LLMs, as evidenced by improved performance in se-

Table 1: Performance comparison on ML-1M and Amazon Review datasets. We highlight the best performance in **bold**. N@$K$ denotes NDCG@$K$.

| Method | ML-1M | | | Games | | | Kindle | | |
|---|---|---|---|---|---|---|---|---|---|
| | N@1 | N@5 | N@10 | N@1 | N@5 | N@10 | N@1 | N@5 | N@10 |
| BM25 | 4.00 | 13.14 | 20.53 | 16.50 | 30.09 | 37.19 | 6.50 | 18.07 | 24.96 |
| UniSRec | 9.00 | 20.08 | 26.72 | 19.50 | 34.86 | 40.82 | 5.00 | 16.21 | 25.03 |
| VQ-Rec | 9.50 | 19.52 | 27.11 | 5.50 | 16.76 | 25.27 | 4.30 | 14.22 | 23.58 |
| Sequential | 21.43 | 42.57 | 48.59 | 24.12 | 47.26 | 53.03 | 10.20 | 27.96 | 33.72 |
| RF | 26.56 | 45.99 | 51.27 | 25.63 | 50.02 | 53.72 | 11.11 | 28.77 | 35.71 |
| ICL | 26.40 | 47.51 | 53.32 | 26.00 | 49.68 | 53.63 | 13.07 | 30.82 | 36.41 |
| Cluster | 27.00 | 45.82 | 52.04 | 26.15 | 47.41 | 52.39 | 13.20 | 25.77 | 34.07 |
| PCL | 29.16 | 48.44 | 54.21 | 29.00 | 51.56 | 55.11 | 11.55 | 29.45 | 36.46 |
| GCL | 30.50 | 48.53 | 53.26 | 32.00 | 51.61 | 56.63 | 10.00 | 31.45 | 36.67 |
| PCL + Cluster | 30.50 | 48.35 | **54.88** | 35.50 | 53.89 | 58.74 | 12.00 | 30.15 | **38.23** |
| Tempura | **31.50** | **48.64** | 54.49 | **39.00** | **56.51** | **60.95** | **14.00** | **32.17** | 37.59 |

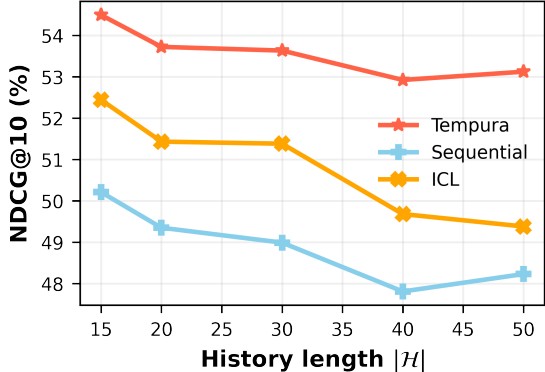

Figure 3: Performance vs. history length $|\mathcal{H}|$ (ML-1M).

Table 2: Performance of Tempura with randomized items, clusters and correctly ordered inputs.

| | Item-R | Cluster-R | Correct |
|---|---|---|---|
| ML-1M | 51.78 | 52.47 | 54.49 |
| Games | 51.83 | 54.18 | 60.95 |
| Kindle | 34.13 | 33.92 | 37.59 |

quential recommendations. We believe that training-based methods essentially learn dataset-specific biases and thus cannot be readily adapted to other domains. Existing study (Hou et al., 2022) shows that training-based methods exhibit significantly reduced performance when applied to domains outside their training scope. Our study aligns with recent research focusing on studying the cross-domain abilities of sequential recommendation models. Accordingly, we compare our method with SOTA cross-domain methods UniSRec and VQ-Rec.

**Implementation details.** Considering economic and efficiency factors, we follow (Hou et al., 2023b; Xu et al., 2024) to randomly sample 200 users along with their historical interactions for each dataset. Unless specified, we use the Azure OpenAI API gpt-3.5-turbo[2]. We set

history length $|\mathcal{H}|$ as 15 and use the most recent 5 interactions as demonstrations in PCL. We found the length of the history significantly affects performance; therefore, we also searched for the optimal $|\mathcal{H}|$ for baselines. Empirically, $|\mathcal{H}| = 10$ yielded the best results for baselines in general. All the reported results are the average of three repeat runs to reduce the effect of randomness.

**Main results.** We present the results on three datasets in Table 1. We can observe our prompting strategies in the third group improves upon existing baselines across all metrics. It is interesting to observe that PCL outperforms ICL significantly, where more demonstrations are used in PCL but ICL only use the last interaction as demonstration. This observation align with our analysis that more demonstrations are needed to learn to utilize temporal information in historical interaction sequences. Although the Cluster strategy exhibits limited performance on its own, it can significantly enhance performance when combined with other strategies in an ensemble. Additionally, we provide a case study of cluster analysis results in Section 4.4. By comparing individual prompting strategies with two ensemble-based methods, we find that ensembling consistently enhances performance

[2]https://azure.microsoft.com/en-us/pricing/details/cognitive-services/

openai-service/

by leveraging the strengths of different strategies. This suggests that different strategies emphasize various aspects, resulting in complementary results.

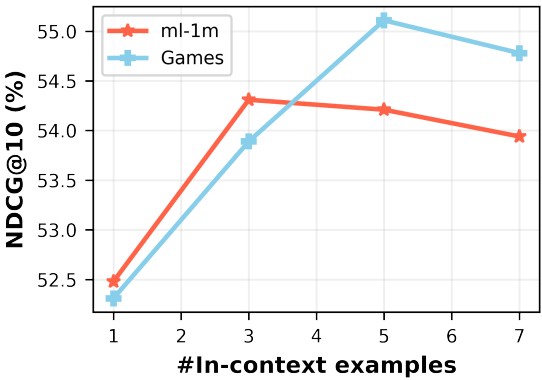

Figure 4: Impact of #in-context examples in PCL. Several more examples can improve performance.

## 4.2. Sensitivity of Temporal Information

In this paper, we aim to improve temporal awareness by designing temporal-aware prompting strategies. To evaluate whether these proposed strategies effectively capture and utilize temporal information within historical interaction sequences, we compare the performance with randomized and correctly ordered histories. We hypothesize that an approach adept at utilizing temporal information should demonstrate superior performance with correctly ordered history. Specifically, our manipulation occurs at two levels: the item level and the cluster analysis level. At the item level, we alter the order of individual items, while at the cluster analysis level, we rearrange the order of clusters derived from cluster analysis. We present the results in Table 2. After randomizing the history, performance on all the datasets drop significantly. This phenomenon indicates that understanding and effectively utilizing temporal information within historical interaction sequences is crucial for capturing and predicting users' future interests.

## 4.3. Ablation Study

**Impact of history length.** It has been reported in Hou et al. (2023b) that increasing the number of historical user behaviors does not improve the ranking performance, but even negatively impacts the ranking performance. To study the impact of history length on Tempura, we vary the history length $|\mathcal{H}|$ used for constructing the prompt from 15 to 50. We compare Tempura with the standard baseline Sequential and the best performing baseline ICL. Here history length $|\mathcal{H}|$ is the maximum allowed history length, the real history length could be shorter. We did not include the results on Games and Kindle since the user interaction history on these two datasets is short.

The results are reported in Figure 3. We observe that utilizing a longer history does not improve performance; in fact, it results in decreased performance on the ML-1M dataset. We hypothesize that the extensive history distracts LLMs, making it difficult for baselines to understand the evolution of user interests. By using temporal-aware prompts and the prompt ensemble strategy, Tempura demonstrates robust performance even with long historical interaction sequences.

**Impact of the number of in-context examples.** We utilize a user's historic items as in-context demonstrations to understand the temporal information in his / her behavior sequence. It is important to understand how many examples are needed. To this end, we study the performance with different number of examples in PCL. We keep the total length of the user's history as 15 and use the latest $k$ items as examples, setting $k$ to values in the set $[1, 3, 5, 7]$. We report the results on the ML-1M and Games datasets in Figure 4. We can observe more examples can boost the performance significantly than only one demonstration. As we analyzed in Section 3.2, LLMs learn to utilize temporal information by learning to predict a series of historical items. However, it is not always the case that more is better. It is observed that a slight performance drop with more examples. We speculate that longer prompts may cause distraction for LLMs.

**Results on GPT-4.** More advanced LLMs, like GPT-4 (Achiam et al., 2023), demonstrate enhanced capabilities in knowledge, understanding, and reasoning. Therefore, we evaluate the sequential recommendation performance using GPT-4 to determine if Tempura can also augment GPT-4's capabilities. Specifically, we apply Tempura to gpt-4-0603. We present the results in Table 4. It has been observed that GPT-4 exhibits a robust capacity for sequential recommendation, even when employing the most standard prompting strategy, Sequential. Notably, the improvement is most significant on the Kindle dataset, leading to the hypothesis that GPT-4 possesses extensive knowledge about Kindle books. The performance improvement with GPT-4 shows its strong ability in understanding and utilizing temporal information. By applying Tempura, the performance can be further improved when the backbone LLM is more powerful.

## 4.4. Case Study

We present an example result from the cluster analysis conducted on the Games dataset. We employ gpt-3.5-turbo to cluster historic items using the prompt discussed in Section 3.2. The historic items was successfully clustered into 4 clusters, accompanied by a generated summary for each cluster. It can be easily observed that the user's most recent interest lies in action shooting games. With this analysis, the target item can be easily identified since it is a first-person action-adventure

Table 3: Case study of structure analysis in the historical interaction sequence.

| |
|---|
| Cluster 1: [Mad Max - PlayStation 4, Metal Gear Solid V: The Phantom Pain - PlayStation 4]. |
| Cluster summary: Action games on PlayStation 4. |
| Cluster 2: [Star Wars: Battlefront - Standard Edition - PlayStation 4, Fallout 4 - PlayStation 4, Just Cause 3 - PlayStation 4, Far Cry Primal - PlayStation 4 Standard Edition]. |
| Cluster summary: Open-world action games on PlayStation 4. |
| Cluster 3: [Tom Clancyś The Division - PlayStation 4, Uncharted 4: A Thiefś End - PlayStation 4, Homefront: The Revolution - PlayStation 4, Deus Ex: Mankind Divided - PlayStation 4]. |
| Cluster summary: Action games with a focus on story and/or multiplayer on PlayStation 4. |
| Cluster 4: [Rise of the Tomb Raider: 20 Year Celebration - PlayStation 4, Dishonored 2 - PlayStation 4, Resident Evil 7: Biohazard - PS4 Digital Code, Horizon Zero Dawn - PlayStation 4, Tom Clancy's Ghost Recon Wildlands - PlayStation 4]. |
| Cluster summary: Single-player action shooting games with a focus on exploration and/or stealth on PS4. |
| **Target item:** Prey - Pre-load - PS4 Digital Code First-person action-adventure shooting game |

Table 4: Performance with GPT-4 (NDCG@10). `Tempura` can further improve the performance when the backbone LLM is more powerful.

| Method | ML-1M | Games | Kindle |
|---|---|---|---|
| Sequential | 55.75 | 66.43 | 57.65 |
| ICL | 54.82 | 67.84 | 54.72 |
| Tempura | 58.39 | 68.13 | 58.59 |

shooting game, aligning with the user's latest interest.

### 4.5. Computational Costs and Latency Analysis

Our `Tempura` is built upon the capabilities of commercial LLMs, such as GPT-3.5 and GPT-4, leveraging their advanced natural language understanding and generation capabilities. The running cost associated with `Tempura` varies to the complexity and length of the descriptive texts used for specific datasets. For instance, in the case of the MovieLens dataset, we use movie titles as descriptive texts, whereas for the Amazon Game dataset, game titles are used. This variance in descriptive text complexity directly influences the computational resources required, thereby affecting the overall cost of running `Tempura`.

To approximate the running cost of a single `Tempura` run, we refer to the API pricing detailed on the OpenAI website[3]. Each `Tempura` execution involves 4 API calls: 3 calls for ranking and 1 call for structure analysis. To provide a sense of the cost of running `Tempura`, we base our cost calculation on the assumption that each of these API calls processes an input of 1,000 tokens and generates an output of 500 tokens. Utilizing the `gpt-3.5-turbo-0125` model, the

price per API call under these conditions is calculated to be $0.00125. Therefore, the total cost for a single recommendation cycle using `Tempura` is $0.005. The latency of a `Tempura` run is directly related to the inference latency of employed LLMs. When using `gpt-3.5-turbo-0125`, it takes around 15s to finish a full `Tempura` run. We will add a more comprehensive computation costs and latency analysis to understand the cost-benefit trade-offs of using `Tempura`. Although latency is not our main focus in this paper, we believe that, with the rapidly evolving inference technologies and the related infrastructure, the latency observed in our current paper can be reduced significantly soon, rendering our framework and other LLM-based recommendation systems more practical for real-world use cases.

## 5. Conclusion

In this paper, we focus on improving the temporal awareness of LLMs through the study of the sequential recommendation problem. Specifically, we introduce two kinds of prompting strategies: one to learn sequential recommendations via in-context learning and another to explicitly analyze the temporal structures in historical interaction sequences. An ensemble strategy is adopted to aggregate results from various prompting strategies. Our study demonstrates that by incorporating specific prompting strategies, LLMs can significantly improve in capturing and utilizing temporal information. This advancement not only strengthens the capabilities of LLMs in sequential recommendation tasks but also opens up new avenues for applying these models in time-sensitive domains.

---

[3]https://openai.com/pricing

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
