# OpenReview forum: "Improve Temporal Awareness of LLMs for Domain-general Sequential Recommendation"
_ICML.cc/2024/Workshop/ICL — ICML 2024 Workshop ICL Poster_

### Official Review · Reviewer_31sz · 2024-06-08
**I suggest accepting this paper based on thorough and well-justified experiments and no major issues with the content.**

**Rating:** 2
**Fit:** 3
**Confidence:** 2

**Workshop Review:**

Large language models (LLMs) have trouble correctly understanding temporal information in their context window. This can lead to reduced performance in domains where understanding the order of temporal or sequential data is important. This is the case when using LLMs to zero-shot rank items for sequential recommender systems (SRS). In fact, the authors show that randomizing the order of sequential information produces near-identical performance in baseline methods, which suggests that these methods don’t fully use sequential information.
The submitted work aims to improve temporal awareness of LLMs by proposing a novel prompting framework. The suggested prompting strategies are justified by findings about LLM behavior from prior work and some results from neuroscience. First, proximal temporal demonstrations teach the model in context how predictions depend causally on history. Second, global interest demonstrations preserve users’ long-term interests despite possible truncation of the history. Lastly, an LLM is prompted to cluster the historical data. Additionally, the results from different prompting strategies are aggregated by aggregating them as an ensemble.
The authors evaluate their method by comparing it to various prompt-based and cross-domain baselines from the literature. The proposed method is shown to both improve the temporal awareness of LLMs and lead to better downstream performance.

—Strengths—

The problem is well justified: Prior work identifies temporal awareness as a blind-spot in LLMs.
The experiments comprehensively analyze the proposed solution: The main experiments compare with numerous baselines. The ablation studies provide interesting context, such as the length of the history and the number of in-context examples.
The paper contributes a useful framework to improve temporal awareness of LLMs without fine-tuning weights. The authors show this improves performance in the SRS domain, and it might be applicable to other domains where temporal awareness is relevant.

—Weaknesses—

The main reason I am recommending this paper as a poster instead of an oral presentation is that only part of the method fully utilizes in-context learning, whereas other parts follow more traditional prompting approaches. Thus, there might be other papers that are a stronger fit for this workshop

—Minor Suggestions—

Though I don’t see this as something that should influence the paper’s acceptance, the reference to neuroscience w.r.t. the ensembling and clustering techniques seem like a bit of a stretch.
A minor improvement would be to provide an explicit example of why the sequential order matters for SRS, which seems to mainly be the changes in the recommender's preferences over time. This isn’t explicitly stated until section 3.2.
Some typos:
- Introduction, first paragraph ‘in deed’ -> ‘indeed’
- Related work, last paragraph: “on a specific domains and can not be easily transfer to unseen domains” -> “on a specific domain and cannot easily transfer to unseen domains”
Should probably change the page titles to not say “Submission and Formatting Instructions for ICML 2024”

I suggest accepting this paper based on thorough and well-justified experiments and no major issues with the content.

**Reason For Not Giving Higher Score:**

The main reason I am recommending this paper as a poster instead of an oral presentation is that only part of the method fully utilizes in-context learning, whereas other parts follow more traditional prompting approaches. Thus, there might be other papers that are a stronger fit for this workshop

**Reason For Not Giving Lower Score:**

The paper does provide novel insights into temporal awareness of in-context data, which is relevant to this workshop. The paper is clear, and I have found no issues with its correctness.

---

### Official Review · Reviewer_8z2J · 2024-06-10

**Rating:** 2
**Fit:** 3
**Confidence:** 2

**Workshop Review:**

The paper introduces a prompting method for improving the temporal prediction ability of LLMs. This prompting method consists of three components: (1) in-context learning module; (2) temporal structure analysis; and (3) prompt ensemble. The experimental results show some improvement with the proposed prompting method.

**Reason For Not Giving Higher Score:**

The method is poorly motivated and the ablation study is missing.

**Reason For Not Giving Lower Score:**

The experimental results show certain advantages of the proposed method.

---

### Meta-Review · Area_Chair_52EN · 2024-06-17

**Recommendation:** 2

**Metareview:**

Reviewers agreed that the evaluation demonstrates the necessity of accounting for lack of use of sequential information by LLMs in in-context learning settings, motivating the proposed technique. I agree with a reviewer that it would be preferable to omit reference to inspiration from "human cognitive processes" and "neuroscience" as these claims are inaccurate (due to missing grounding in the science of human cognition) and superfluous (unnecessary to support the main claims of the paper).

---

### Decision · Program_Chairs · 2024-06-17

**Decision:**

Accept (Poster)

**Comment:**

**Accept with minor revision**: Please revise references to neuroscience and cognitive science to be appropriate for the scope and contributions of the paper.